# Nutritional Content of Sliced Bread Available in Quebec, Canada: Focus on Sodium and Fibre Content

**DOI:** 10.3390/nu13124196

**Published:** 2021-11-23

**Authors:** Marie Le Bouthillier, Julie Perron, Sonia Pomerleau, Pierre Gagnon, Marie-Ève Labonté, Céline Plante, Marc-Henri Guével, Véronique Provencher

**Affiliations:** 1Centre NUTRISS—Nutrition, Santé et Société, Institut sur la Nutrition et les Aliments Fonctionnels (INAF), Université Laval, Quebec, QC G1V 0A6, Canada; marie.le-bouthillier.1@ulaval.ca (M.L.B.); julie.perron@fsaa.ulaval.ca (J.P.); sonia.pomerleau@fsaa.ulaval.ca (S.P.); pierre.gagnon.10@ulaval.ca (P.G.); Marie-Eve.Labonte@fsaa.ulaval.ca (M.-È.L.); 2École de Nutrition, Université Laval, Quebec, QC G1V 0A6, Canada; 3Institut National de Santé Publique du Québec, Quebec, QC G1V 5B3, Canada; celine.plante@inspq.qc.ca; 4Faculté des Sciences de l’agriculture et de l’alimentation, Université Laval, Quebec, QC G1V 0A6, Canada; Marc-Henri.Guevel@fsaa.ulaval.ca

**Keywords:** bread, nutritional content, monitoring, sodium, fibre, front-of-pack (FOP), observatory, food offer, food purchases

## Abstract

Bread is a highly consumed food and an important source of nutrients in the diet of Canadians, underlining the need to improve its nutritional quality. The Food Quality Observatory (Observatory) aimed to evaluate the nutritional value of the sliced bread products available in Quebec (Canada), according to their grain type and main flour. Analyses included comparing the bread’s sodium content to Health Canada’s daily values (DV) and sodium voluntary targets, comparing the bread’s fibre content thresholds with the source of fibre mentioned, and assessing whether the main flour was associated with the nutrient content. The nutritional values of 294 sliced breads available in Quebec were merged with sales data (from October 2016 to October 2017), with 262 products successfully cross-referenced. The results showed that 64% of the breads purchased were ‘low’ in fibre (i.e., below 2 g per two slices), while 47% were ‘high’ in sodium (i.e., above 15% DV per two slices). Attention should be paid to 100% refined grain bread and to bread with refined flour as the main flour, since they are great sellers, while showing a less favourable nutrient content. This study shows that there is room for improvement in the nutritional content of Quebec’s bread offer, particularly regarding sodium and fibre content.

## 1. Introduction

Highly processed foods, which are processed or prepared foods that high are in sugar, saturated fats, and sodium [1], are widely consumed by individuals [2]. The nutritional content of processed foods has the potential to be improved, for the benefit of individuals’ diets and ultimately population health [3]. The World Health Organization (WHO) has, indeed, prioritized a global goal of reducing salt consumption by 30% by 2025 [4]. Such an initiative toward sodium reduction has been recognized as a very cost-effective public health strategy [5], considering that high sodium intake is associated with high blood pressure, fatal coronary heart disease, risk of stroke, and organ damage [6,7]. Adequate fibre intake is also an important component of a healthy diet, as diets high in fibre are linked to a lower risk of developing cardiovascular diseases, diabetes, obesity, and certain gastrointestinal diseases [8,9]. Sodium and fibre therefore represent two nutrients that offer an appropriate starting point for food reformulation in processed foods.

One way to initiate nutritional improvements is to collaborate with agri-food industries and support them in ensuring a consistency between public health goals and the nutritional quality of the food products they offer to consumers. Notably, to reduce the availability of foods high in sodium on the market, the WHO suggests working with the private sector [4]. In order to have a sustainable and healthy food system, the American Heart Association also recommends that public health authorities collaborate with industry [10]. As many as 80% of countries around the world are targeting agri-food industries in their national strategies to reduce sodium consumption [11]. More specifically, about 40 countries have adopted governmental targets for sodium reduction in foods, with three quarters using voluntary targets and one quarter implementing mandatory regulations [12,13]. Regarding fibre initiatives, to our knowledge, no studies have yet reported national programs or regulations directed at agri-food industries to increase the content of fibre in their products or the impact that this could directly have on the food offered, and ultimately on health. Nonetheless, conclusions similar to those for sodium could possibly be drawn with respect to fibres potential health effect on the population [14]. Therefore, collaborations with the agri-food industry appear to be a relevant strategy to improve sodium intakes and an innovative strategy to improve the fibre contents of processed foods.

In terms of food products that should be specifically targeted, sliced bread is an ideal staple food to focus on, since it is consumed by all socio-economic classes of the population [15]. By sliced bread, we mean a loaf of bread that has been pre-sliced by machine and packaged in that form at the processing plant, and whose packaging includes a nutrition facts table and unique product code (UPC). This excludes artisanal bread or bread sliced at the point of purchase. Bread, in general, provides a high amount of sodium and a low amount of fibre in the diet. In Quebec (Canada) particularly, bread has an important impact on the population’s sodium intake, representing 11.3% of the total daily sodium intake for adults, and being among the largest contributors of sodium due to its frequent consumption [16]; it represents over a quarter (27%) of all cereal products consumed [17]. Bread consumption also plays a role in the daily fibre intake, although the fibre content of bread could be further increased. Indeed, bread contributes to 14% of Quebecers’ fibre intake, with 64% of the bread consumed being made mainly with refined grains and 36% being made mainly with whole grains [17]. In terms of intake, both the sodium and fibre consumptions of Quebecers do not meet the recommended health targets [16,17,18,19]. According to a modeling conducted by the Institut national de santé publique du Québec (INSPQ), a sodium reduction equivalent to one standard deviation of the sodium content in bread Quebecers’ purchase in stores would reduce sodium dietary intake by at least 65 mg/day/person (or −7.5% of the sodium content of their food basket) [20]. Bread reformulation, thus, seems to be a relevant method to improve nutrient intakes, not only in Quebec, but also for other populations.

Although it appears to have great potential to improve sodium and fibre intakes through collaboration with sliced bread’s agri-food industries, no studies have yet systematically monitored the extent to which these products could be improved, particularly in Quebec, Canada. One way to identify the products that could be enhanced, in terms of fibre and sodium, is to compare their content to the daily value (DV) recommended by Health Canada, which was 28 g for fibre and 2300 mg for sodium in 2016–2017 (time of data collection) [21]. This information, displayed to consumers in the nutrition facts table, indicates whether a product contains a lot or a little of a given nutrient by portion: 5% of the DV or less being considered a little, 15% of the DV or more being a lot of that nutrient. For sodium, it should ideally be less than 15% and for fibre, more than 15%. However, for fibre, we chose thresholds of 2 g and 4 g of fibre for our analysis (which correspond to 8% and 16% of the DV). These are, respectively, the quantities necessary to have the claims ‘source of fibre’ and ‘high source of fibre’ on the front-of-package (FOP) of products. Little data are available on to what extent sliced bread is reaching, or not, these sodium and fibre thresholds in Canada.

In 2012, Health Canada published voluntary sodium reduction targets for processed foods, to be implemented by the end of 2016 by agri-food industries [22]. These targets differ from the 15% DV threshold, are adapted to the food matrix, and are intended to encourage a gradual reduction (in phases), while continuing to maintain food safety, quality, and consumer acceptance. Furthermore, the targets in each phase were developed in consultation with the agri-food industry, the health sector, and research experts. The ultimate target for bread, the phase III (330 mg/100 g for most bread, except 260 mg/100 g for raisin bread) would result in the average sodium content in bread being around 10% of the Health Canada DV (247 mg/75 g, approximately 2 slices). Five years after the introduction of the first voluntary targets for sodium reduction in processed foods, Health Canada reported that 48% of all products included in their study (including bread) had not significantly reduced, and that some products even increased, their sodium content. For sliced bread, the average sodium content only reached the first stage of reduction (an average of 424 mg sodium/100 g) and a significant percentage of breads were still over the maximal level of sodium target for bread (520 mg/100 g or 390 mg per portion of 75 g). These results suggest that further work should be done to reduce the sodium content of the sliced bread offered in Canada [23]. 

In this context, the current study aimed to provide some insights into the concerns raised above by following three objectives: (1) To characterize the nutritional content and selling price per portion of the sliced bread offered and purchased in the province of Quebec (Canada), according to their type of grain; (2) To verify, according to the grain type, the distribution of sliced bread’s sodium and fibre content based on (a) percentage DVs for sodium and 2 g and 4 g for fibre and (b) Health Canada 2012 voluntary targets for sodium; and (3) To characterize the associations of the sliced bread’s main flour with certain nutrients of interest the selling price. As part of the third objective, we also assessed whether controlling for other information on the packaging could influence the observed associations. Overall, the results from this project will provide details on areas for improvement in this food category, as well as indications for future actions targeting product labels and knowledge mobilization.

## 2. Materials and Methods

### 2.1. Data Collection 

The methodology used to assess the nutritional content of the food offered was developed by the Food Quality Observatory [24]. In order to reach the objectives described above, a database containing the nutritional value of each sliced bread offered on the market was created by Protégez-vous [25]; a Quebec-based non-profit organization specializing in consumer information and product testing. This database was used by the Observatory through a data-sharing agreement. Protégez-vous collected nutritional and labeling information by purchasing every sliced bread sold in supermarkets (e.g., Metro, IGA, Provigo), wholesale stores (e.g., Walmart, Costco), or specialty grocery stores (e.g., Avril, Rachelle-Béry) between December 2016 and March 2017 in the greater Montreal area, Quebec, Canada. Types of bread included in the current study were only the sliced breads that were prepackaged and displaying a nutrition facts table. All other types of bread (e.g., baguette, tortillas, etc.) and sliced bread without a unique product code (UPC) were excluded from this study. A reference portion of two slices of bread (or at least 50 g, if two slices corresponded to less than 50 g) was chosen because it represented Health Canada’s reference amount at the time of the study [26]. Nutritional value variables listed for the purpose of this study were as follows: energy (kcal), lipids (g), saturated fats (g), sugar (g), carbohydrates (g), fibre (g), protein (g), and sodium (mg). The price per reference serving ($) was also obtained from this database, by calculating the average of the prices observed in the various stores visited. This database of nutritional values was merged with a sales database (provided by the Nielsen company [27]) based on UPC. The sales database included all bread sales in Quebec gathered from principal supermarket sales (e.g., Loblaws, Sobeys, Metro, Walmart) for a 52-week period, ending October 2017. For each product, the Nielsen database included sales in kilograms (kg).

### 2.2. Classification

Sliced bread products were analyzed according to two classifications namely the type of grain used in the bread (e.g., single grain 100% refined, multigrain 100% whole) and the main flour used (e.g., whole wheat, refined wheat, rice) according to the list of ingredients (see Table 1).

In the classification based on the type of grain, detailed in Table 1, the term grain refers to both cereal grain (e.g., wheat, kamut, barley) and seeds/oilseeds (e.g., soy, sesame, flax). The term grain also refers to both flours and added grains. Types of bread were classified according to whether they contained only one type of grain or two or more types of grain (i.e., single vs multigrain, respectively) and whether they were 100% whole grain, 100% refined grain, or mixed grain (i.e., a combination of whole and refined grains). It should be noted that there is no regulation in Canada regarding the use of the term ‘whole grain’ for bread [28]. Thus, although some products are labelled as ‘whole grain’, many contain negligible amounts of whole grain. It should be noted that ingredients listed under ‘topping’ were not considered in determining the type of grain. In addition, all starches (potato starch, corn starch, etc.), gluten, as well as all fibres (oat husk fibre, inulin, etc.) were not considered in this classification.

### 2.3. Statistical Analysis

To provide a general description of the nutritional content and selling price of sliced bread offered in Quebec (i.e., bread supply), means and standard deviations were calculated for all types of bread and for every grain type (objective 1). These descriptive analyses were then repeated by weighting according to sales volume (i.e., bread purchases). Since analyses of purchases were made from the merged database, the number of sliced bread products was lower (*n* = 262) than when using the nutritional database only (*n* = 294), because some products could not be successfully cross-referenced. 

To show how the sodium content of sliced bread is distributed in relation to Health Canada’s percentage daily value (DV) 2012 voluntary reduction targets for sodium and the threshold for fibre claims, bubble plots were created (objective 2). They showed simultaneously, for every grain type, how many products and what volume of sales exceeded the thresholds. In terms of limits, the intervals ‘less than 5%’ for sodium and ‘less than 2 g’ for fibre, ‘between 5% inclusive and 15% exclusive’ for sodium and ‘between 2 g inclusive and 4 g exclusive’ for fibre, and ‘15% and above’ for sodium and ‘4 g and above’ for fibre were used.

To characterize the associations between the main flour and nutritional content, as well as selling price (objective 3), Kruskal–Wallis tests were used, since the residuals were not normally distributed while using standard ANOVA. These analyses were also repeated by weighting for sales volume. Finally, rank regression was used to assess whether associations observed with the Kruskall–Wallis tests were similar when adjusted for the type of grain and other product information, including organic claims, mention of being natural or rustic, gluten-free claims, and market segments.

All statistical analyses were performed using SAS 9.4 TS1M4. The code can be obtained from the corresponding author upon request.

## 3. Results

A total of 294 different types of sliced bread were identified in the Quebec food supply from the data collection by Protégez-vous. Nutritional values and all packaging information were referenced for these types of bread. Using the UPC, this dataset was then merged with the sales database from Nielsen, which comprised 1384 types of bread sold over one year. The higher number of breads in this dataset was attributable to two factors: [1] Nielsen listed a larger number of small vendors, which Protégez-vous did not capture, and [2] Nielsen’s file covered a larger market than Protégez-vous (i.e., including other types of bread, such as unsliced bread, as opposed to sliced bread only). A total of 262 products with sales information were successfully cross-referenced with the 294 sliced breads identified by Protégez-Vous. Those 262 products represented 413 million dollars, 85 million kg, or 75% of the sales volume of all sliced bread sold in Quebec. Considering that the sales volume in Canada is CAD 3.5 billion for all bread, of which CAD 2.2 billion comes from sliced bread (62% of all bread) [29], we estimated that the coverage of 75% of all bread products is representative of the Quebec sliced bread market.

First, to describe the samples of sliced bread under study, Table 1 shows the repartition of the variety of products and of sales volume of sliced bread according to the grain type and main flour, and with definitions of the categories. The most represented variety of sliced bread, based on grain type, were multigrain mixed grains followed by multigrain 100% whole grains. In terms of the main flour, bread products with refined wheat flour were the most represented, followed by whole wheat flour. Multigrain mixed grain bread products were the most frequently purchased, followed by 100% refined grain bread. In the classification based on main flour, bread products with refined wheat flour as the main flour were by far the most purchased, while bread products with whole wheat flour as the main flour were second.

**Table 1 nutrients-13-04196-t001:** Offer, purchases, and definitions of sliced bread according to their types of grain and main flour.

Classifications	Categories	Definitions	Offer ^a^ (*n* (%))	Purchases ^b^ (% of Sales Volumes)
Type of grain	100% whole grain	Single-grain bread made from 100% whole grain (e.g., whole wheat flour bread).	39 (13.3)	12.2
	100% refined grain	Single-grain bread made from 100% refined grain (e.g., enriched wheat flour bread (white)).	22 (7.5)	24.5
	Mixed grain	Bread made of a single type of grain, partly whole grain and partly refined grain (e.g., whole wheat and fortified flour bread).	14 (4.8)	1.0
	Multigrain 100% whole grain	Bread made of two or more grain types and 100% whole grain (e.g., whole wheat flour and whole spelt flour bread).	54 (18.4)	5.8
	Multigrain 100% refined grain	Bread made of two or more grain types and 100% refined grain (e.g., whole wheat flour and whole spelt flour bread).	9 (3.1)	1.2
	Multigrain mixed grain	Bread made of two or more types of grain and composed of whole grain and refined grain (e.g., buckwheat flour and fortified wheat flour bread).	134 (45.6)	52.6
	Raisin bread (or other fruits/vegetables) 100% refined	Fruit or vegetable bread made from 100% refined grain (e.g., raisin bread with fortified wheat).	5 (1.7)	2.4
	Raisin bread (or other fruits/vegetables) mixed grain or whole grain	Fruit and vegetable breads that are not solely composed of refined grain (e.g., cranberries bread with enriched and whole wheat).	17 (5.8)	0.4
Main flour	Whole wheat flour	Whole wheat flour, wholemeal flour, whole wheat flour with germ, sprouted wheat flour, wholemeal flour.	103 (35.0)	28.3
	Refined wheat flour	White flour, enriched flour, wheat flour, enriched wheat flour, unbleached flour.	114 (38.8)	70.6
	Rice flour	Rice flour, brown rice flour.	43 (14.6)	0.7
	Other flours	Flour other than those listed above (quinoa, spelt, kamut, rye, etc.).	34 (11.6)	0.4

^a^*n* total = 294; ^b^
*n* total = 262.

Table 2 shows that the nutritional value and price per portion of the offered and purchased sliced bread, based on the type of grain. Offer represents the average nutritional composition of the bread offered on the shelves (*n* = 294), while purchases represent the nutritional composition of the bread purchased by consumers (*n* = 262) (nutritional composition weighted by sales volume). In terms of the products offered, the main results show that bread products in the multigrain 100% whole grain category were higher in lipids and had a higher selling price per portion compared to other breads.

Bread offered in the 100% refined grain category contained less fibre and had a lower selling price per portion compared to other bread. While few differences were observed for the bread offered, the main results showed that significant differences were observed when considering the bread purchased. More specifically, bread products purchased in the 100% whole grain category were higher in fibre compared to other types of bread, while purchased bread with 100% refined grain had lower sugar and fibre content, a higher sodium content, and a lower selling price. Purchased multigrain bread with 100% whole grain also differed from other types of bread, with a higher content in lipids, fibre, and protein and a higher selling price. Purchased raisin bread products with 100% refined grain were higher in sugar and saturated fats compared to the others,

Figure 1 shows the large variations in sodium content within each bread classification, based on their type of grain. In terms of the products offered (i.e., not adjusted by sales), 26% of breads are above the 15% DV for sodium. By illustrating the sales for each product, this figure shows that many important sellers were high in sodium, and almost half of them exceeded 15% (350 mg) of the DV for sodium. Indeed, 47% of breads purchased in Quebec were above the 15% DV cut-off recommended by Health Canada in terms of sodium content. More specifically, 79% of 100% refined single grain bread and 77% of 100% whole grain bread products purchased were above this cut-off. 

Figure 2 shows the sliced bread’s content in sodium when compared to the voluntary sodium reduction targets for the third phase proposed by Health Canada in 2012 (330 mg or 260 mg per 100 g of bread). The results showed that a high proportion of all bread offered (i.e., 77% of products) still exceeded the phase III and ultimate targets five years after their implementation by Health Canada. More specifically, all 100% refined grain bread, 95% of raisin bread with 100% refined grain, 90% of the bread with 100% whole grain, and 89% of multigrain bread with 100% refined grain exceeded the voluntary reduction targets for sodium. When considering the products purchased, 87% exceeded the targets, including all 100% refined grain bread, multigrain bread with 100% refined grain, raisin bread with 100% refined grain, as well as 98% of raisin bread with mixed or whole grains.

Figure 3 shows the large variation in fibre content observed between the different types of bread, based on their types of grain, as well as within each classification. In terms of the products offered, 34% of all products were below the 2 g for fibre. More specifically, 90% of 100% refined grain bread, 89% of multigrain with 100% refined grain bread, and 100% of raisin bread with 100% refined grains were below the 2 g for fibre. This figure also illustrates the sales for each product and shows that many important sellers were low in fibre, with more than half of them below 2 g of fibre. Indeed, the main results show that 64% of breads purchased were below this 2 g cut-off, meaning they cannot mention ‘source of fibre’ on their FOP. More specifically, 98% of multigrain with 100% refined grain bread, 98% of raisin bread with 100% refined grain, and 95% of 100% refined grain breads purchased were below this cut-off. Only 17% of all breads purchased were above the 4 g cut-off (threshold for mentioning ‘high source of fibre’), with 73% of multigrain 100% whole grain bread and 38% of 100% whole grain bread. 

Table 3 shows the nutritional value and price per portion of sliced bread offered and purchased, based on the main type of flour. In terms of the products offered, bread products with refined flour were lower in fibre and protein compared to bread with whole wheat flour as the main flour. Bread offered with rice flour as the main flour was higher in lipids and sugar, and lower in fibre and protein, while having a higher selling price, compared to bread with whole wheat flour. More important differences were observed when considering the bread purchased. Purchased bread products with refined flour as the main flour were, indeed, lower in fibre, protein, and lipids; higher in carbohydrates and sugar; and sold at a lower price than bread with whole wheat flour as the main flour. Purchased bread products with rice flour as the main flour were again higher in lipids, lower in protein, and sold at a higher price than bread with whole wheat flour.

Table 4 shows a multivariate analysis for the nutritional composition and the selling price of bread according to the main flour, while being controlled for other main specific characteristics of the bread (types of grain, organic claims, natural or rustic mention, gluten-free claims, and market segments) and weighted by sales volume. The main results show that, while controlling for other specific characteristics, bread products made from refined wheat flour were lower in lipids, fibre, and protein, and higher in sugar, carbohydrates, and sodium, while being less expensive than bread made from whole wheat flour.

## 4. Discussion

This study provides a global picture of the nutritional content of the sliced bread offered and purchased in Quebec, according to their grain type and main flour. First, we showed that the greatest variety of products is found in the category of multigrain mixed grain bread, and that most breads offered on the market have refined wheat flour as the main flour. Purchases follow a similar pattern, with multigrain mixed grain bread and bread made with refined wheat flour representing the biggest percentage of sales volume. Notably, although 100% refined grain bread had a low diversity (only 7.5% of all products offered), they were ranked second in terms of sales volume (25% of annual sales). When considering the representativeness of the sample, this study showed a number of products similar to those observed in the greater Toronto area, in Canada (*n* = 301) [28], but a greater variety of products than a study previously conducted in France (*n* = 95 artisanal bread) [30].

Based on the classification according to the type of grain, analyses of the nutritional value showed that, as expected, the fibre content of the offered 100% refined grain bread was lower than in other types of bread. The offered multigrain 100% whole grain bread had a higher lipids content than other bread. This result can be explained by the fact that whole grain includes the germ portion of the grain, which is rich in lipids, particularly mono- and polyunsaturated fats [31]. Differences were mainly observed when comparing the nutritional content of the bread purchased versus that offered. The main results presented in this study showed that purchased breads from the category of 100% refined grain were substantially cheaper, although they were significantly higher in sodium and lower in fibre, compared to other breads. Indeed, most of the possible areas of improvement in sliced bread relate to this category. On the consumer side, a lower purchase of wholegrain foods could be explained by a higher price, as observed for whole grain bread, together with a lack of availability, a dislike of taste, negative physical effects after consumption, and incompatibility with meal patterns [32]. In the interest of considering equity among those who might purchase bread with refined flour and grain because of limited financial resources, reformulation of these products should not result in an increase in their price.

In terms of the sodium and fibre contents, when compared to Health Canada thresholds, 100% refined grain bread reached the 2 g for fibre least often and exceeded the 15% DV for sodium most often. On the other hand, multigrain 100% whole grain bread mostly reached the minimum threshold for 2 g for fibre, while almost all the products were between 5 and 15% DV for sodium. The biggest sellers across all classifications, and particularly bread in the 100% refined grain category, showed a great room for improvement for both nutrients. Considering that many bread products have between 350 mg and 360 mg of sodium, a decrease of only 10 mg of sodium in these bread products could have an important impact on their nutritional quality, because it would make them fall below the 15% threshold. Moreover, since these bread products represent almost 20% of sales, a decrease of 10 mg of sodium of these big sellers could also have an important impact on the population’s sodium intake. Thus, a perfect reformulation would not only reduce sodium below 15% DV and increase fibres above 2 g, but also do so in a meaningful, population-level, manner.

Previous studies have suggested that purchased bread accounts for 5% of Quebecers’ sodium intake (128.5 mg per day per capita out of a mean intake of 2760 mg) [33] and 6% of the fibre intake (1.1 g per day per capita out of a mean intake of 18.1 g) [34]. However, it is important to underline that the sodium content influences all steps of the bread-making process, as well as the final product (bread volume and sensory characteristic) [35]. Fibre content can also influence the overall sensory properties of bread and the bread-making process [36]. These major roles are often cited by the agri-food industries as reasons for not changing the sodium and fibre content of baked goods. However, there are alternatives to reducing sodium and increasing fibre in bread, such as the use of other molecules with similar binding and organic properties or by-products rich in fibres from other industries, but these involve changes in the production, distribution chain, and consumer behavior [35,36,37]. In addition, this analysis showed that some bread products meet the DVs, suggesting that food companies could draw on recipes from these types of bread to reformulate their products.

In 2012, Health Canada proposed voluntary sodium reduction targets for processed foods to encourage industry to reduce sodium in their products in three stages [23]. Health Canada notes that in 2017, the average sodium content was 424 mg per 100 g of bread (sales weighted average), meeting the phase I target of 430 mg. However, with an average of 453 mg of sodium per 100 g of bread purchased (data not shown), the bread products analyzed in this study did not meet the phase I target. The average sodium content of bread offered in Quebec is 403 mg per 100 g of bread (data not shown), while in comparison the content is 455 mg in the United States and 406 mg in the United Kingdom (not weighted with sales average) [38]. Five years after Health Canada proposed voluntary sodium reduction targets, 77% of sliced bread available in Quebec still exceeded the proposed phase III target for sodium (330 mg per 100 g serving of bread). In fact, almost half of the bread offered in Quebec, according to our study, still exceeded the first phase, set at 430 mg per 100 g of bread. In 2020, Health Canada revised these targets and modified them for white bread (refined flour) (360 mg/100 g) and raisin bread (330 mg/100 g), while the target remained the same for whole grain bread and multigrain bread (330 mg/100 g) (weighted with sales average) [39]. However, most types of bread (*n* = 213 or 72%) in the present study would not meet these actualized targets.

In terms of perspectives, it is interesting to note the intention of Health Canada to use the DV as a FOP warning symbol to identify food products high in sugar, sodium, and saturated fat [40]. According to our study, 26% of sliced bread products offered in Quebec would be required to add a ‘high sodium’ warning symbol on their front packaging in the upcoming years if the bill is passed (the publication of the final regulation by the government of Canada is still pending [41]). Despite the fact that voluntary programs for industries may work in a certain extent in reducing the salt content of staple foods, coordinated action with regulations, such as FOP labeling, could have a greater influence on industries’ actions [11,42]. Labeling on the front of products, indeed, seems to influence, to a certain extent, the food industry into reformulating existing products and developing new healthier products [43,44,45]. In Chile, after the introduction of a law requiring FOP labeling of products ‘high in’ energy, total sugar, saturated fat, or sodium, a study found that companies reformulated foods accordingly, with the sodium content of flour-based products (including bread) being significantly reduced [46]. Moreover, a systematic review of ‘real-life’ studies highlighted that consumers are likely to purchase healthier food when there are FOP labels [47]. Based on a meta-analysis, it appears that consumers accept bread with a 40% sodium reduction or less [48]. This same study also suggests that consumer acceptance and the level of sodium reduction follow a linear trend, with consumer acceptance gradually decreasing as the levels of sodium reduction increase [48]. As far as fibres are concerned, consumer demand is likely to increase in the coming years, as healthy products are increasingly in demand [49]. With the microbiota gaining interest in the scientific community and in the population, fibre is likely to be in the spotlight, as it contributes to gut health [50]. Thus, increasing its content or using fibre to improve the nutritional profile of foods will probably be of interest to the food industry, as they could use claims on products’ FOP.

Based on the classification according to the main flour, the main areas for improvement were found when analyzing the nutritional content of the bread purchased. In this regard, analyses showed that purchased bread products with refined wheat flour as the main flour provide less fibre, lipids, and protein; and more sugar and carbohydrates; as well as being less expensive, than bread with whole wheat flour as the main flour. Some explanations on the industry side as to why whole wheat flour is not used more frequently as the main flour could include that breads with whole grains have a shorter shelf life, a higher risk of rancidity, and poorer organoleptic properties [51]. Although the mandatory fortification of white (i.e., refined) flour can improve the nutritional quality of these types of bread in given micronutrients (e.g., folic acid), such nutritional information is not systematically indicated in the nutrition facts table, which was a limit for us in our analyses. Intakes of these micronutrients may also be a lesser public health concern than the consumption of fibre and sodium. In that context, nutritional improvements could be considered in the formulation of the most purchased bread with refined wheat flour, with respect to the list of ingredients. Increasing the proportion of whole wheat flour and decreasing the proportion of refined flour in these types of bread could increase the offer of products with whole wheat flour as the main ingredient, providing more fibre, together with other nutrients, such as vitamins and minerals.

Although of interest from a descriptive perspective, the results based on the classification of the main flour are from univariate analyses that did not control for all confounding factors. Therefore, a series of additional analyses were conducted to control for other specific variables, while weighting by sales volume, in the same multivariate model for the main flour. This additional analysis showed that, based on the available specific information, the main type of flour appears to be an indicator of the nutritional quality of purchased bread, since refined wheat flour types of bread seem to have the greatest room for improvement. Given that the main flour and other specific information considered in this analysis are available on the package (e.g., list of ingredients), these results highlight the importance of considering package information as an indicator of nutritional quality and as a possible indicator for consumers. Future analysis could further investigate how the ingredient list could provide insights into the nutritional quality of bread products and help consumers make healthier choices. For example, it would be interesting to know which other ingredients, besides flour, are also correlated with better nutritional quality and where they should be placed in the ingredient list; and the opposite (which ingredients are an indicator of a less interesting nutritional content).

While demonstrating strengths, our study also presents some limitations, including the fact that the offer of bread can change rapidly, even within a year, and that our methodology captured a picture of the bread offer at a specific moment in time. In addition, our data do not include products without UPC, which may be more artisanal and, therefore, perhaps using sourdough, more whole grain flour, and fewer additives, and, thus, representing less highly processed products. In terms of strengths, this study coupled nutritional data with sales data, which proved to be relevant to gain insight into population consumption. In addition, for the first time, this study presented results for the mixed grain types of bread that are not 100% refined nor 100% whole grain, to distinguish them from other types of bread, which will allow us to follow their evolution over time. The Food Quality Observatory aims to monitor the evolution of the food supply, and hence to replicate the same study within the subsequent years, to follow up on these results.

In conclusion, this study shed some light on the necessity to better monitor the nutritional content of sliced bread products, with an emphasis on the bread purchased, as well as sodium and fibre content. Our analyses proved to be relevant, because they showed an important nutritional quality variability in the bread offered in Quebec, which confirms its potential for nutritional improvement. This is particularly relevant for the sodium and fibre content of bread from the 100% refined grain category, and for bread products with refined flour as the main flour, which are great sellers. As mentioned before, the voluntary sodium targets developed by Health Canada for processed foods are achievable, since they were developed in consultation with the agri-food industry, health sector, and research experts, and also take into consideration consumer and technological barriers, as well as public health objectives. An improved knowledge mobilization between industries, research experts, and the public health sector may potentialize positive effects of monitoring the nutritional quality of food products and of such collaboration [52,53]. Indeed, this first portrait of the nutritional content of bread could help inform the industry of potential areas of improvements. The next monitoring of the bread sold in the province will allow estimating if any improvements occur after changes in regulations are put in place (e.g., FOP). Furthermore, product reformulation challenges on the side of the food industry could be better addressed and proactively overcome, for the benefit of public health. In particular, nutritional improvements should be made without favouring one nutrient over another (e.g., decreasing sodium and increasing sugar) or reducing affordability for lower socioeconomic classes. Consumer education could also be an avenue to explore, as it is mainly the most frequently purchased bread which would benefit the most from being improved nutritionally. As a possible solution for all parties concerned, even a modest improvement in the nutritional quality of a staple food such as the bread, which would not harm sales for the agri-food industries or the properties of the bread for consumers, could be a win-win solution for all.

The full report is freely available (in French only) on www.foodoffer.ca (accessed on 20 November 2021).

## Figures and Tables

**Figure 1 nutrients-13-04196-f001:**
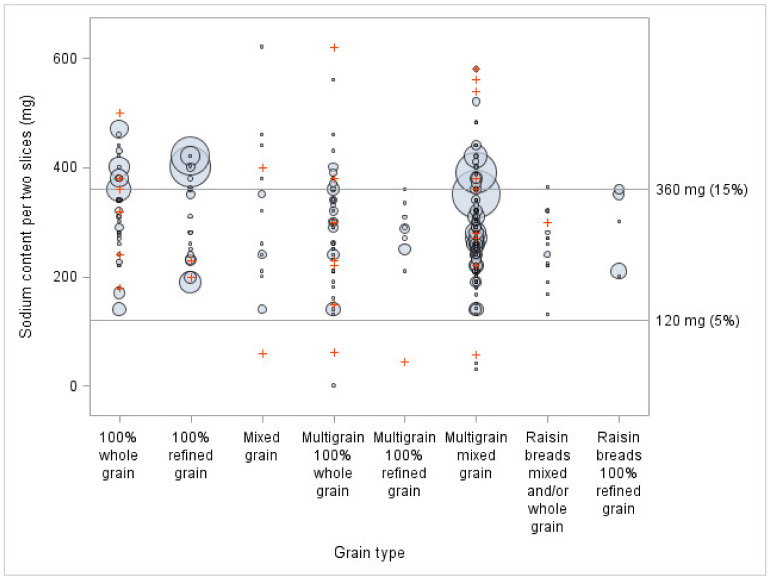
Sodium content per two slices of different breads, by type of grain and sales volume (*n* = 294), compared to Health Canada’s percent daily values (DV). The bigger the circles, the higher the sales. Signs + represent sliced bread for which sales data were not available. A portion of 50 g was used when the weight of two slices was inferior to 50 g.

**Figure 2 nutrients-13-04196-f002:**
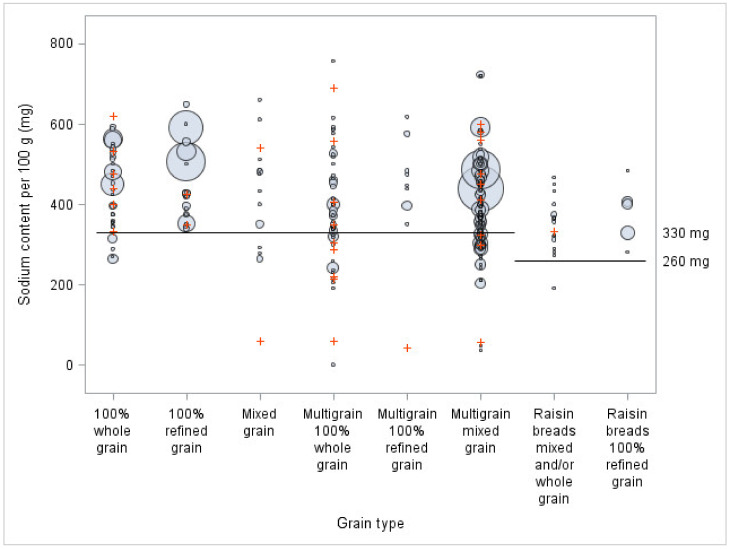
Sodium content per 100 g of different breads, by type of grain and sales volume (*n* = 294), compared to Health Canada final voluntary sodium reduction targets. The bigger the circles, the higher the sales. Signs + represent sliced bread for which sales data were not available. A portion of 50 g was used when the weight of two slices was inferior to 50 g.

**Figure 3 nutrients-13-04196-f003:**
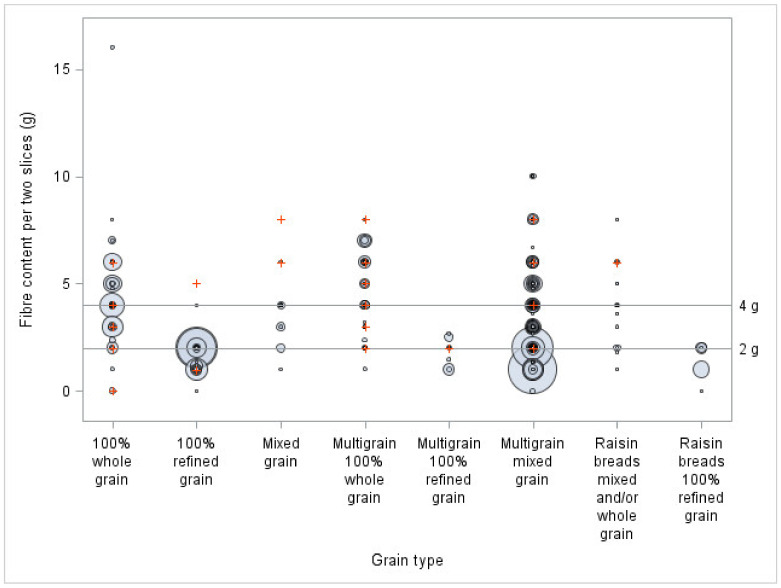
Fibre content of different types of bread and by types of grain and sales volume per two slices (*n* = 294), compared to the 2 g and 4 g threshold (8% and 16% of Health Canada DV). The bigger the circles, the higher the sales. Signs + represent sliced bread for which sales data were not available. A portion of 50 g was used when a two-slice portion was inferior to 50 g.

**Table 2 nutrients-13-04196-t002:** Nutritional value and selling price per two-slice portion of sliced bread offered and purchased in Quebec, according to the type of grain.

	Weight for 2 Slices or min 50 g		Energy (kcal)		Lipids (g)		Saturated Fats (g)		Carbohydrates (g)	
	Offer	Purchases	Offer	Purchases	Offer	Purchases	Offer	Purchases	Offer	Purchases
Total (*n*=294/*n*=262)	73 ± 15	73 ± 12	188 ± 45	189 ± 34	2.9 ± 1.8	2.3 ± 0.9	0.5 ± 0.3	0.5 ± 0.2	35 ± 10	35 ± 7
100% whole grain (*n* = 39)	73 ± 14	74 ± 11	176 ± 41	176 ± 28	2.5 ± 1.7	2.3 ± 0.8	0.5 ± 0.4	0.6 ± 0.3	33 ± 8	32 ± 5
100% refined grain (*n* = 22)	73 ± 13	71 ± 10	185 ± 54	195 ± 34	2.5 ± 1.4	2.2 ± 0.5	0.5 ± 0.3	0.5 ± 0.2	35 ± 10	37 ± 7
Mixed Grain (*n* = 14)	72 ± 17	65 ± 21	201 ± 63	163 ± 66	2.2 ± 2.2	1.3 ± 1.3	0.3 ± 0.2	0.2 ± 0.2	37 ± 11	32 ± 13
Multigrain 100% whole grain (*n* = 54)	70 ± 19	76 ± 14	187 ± 41	190 ± 45	4.1 ± 2.3 *	3.1 ± 1.6 *	0.5 ± 0.4	0.6 ± 0.3	34 ± 10	34 ± 7
Multigrain 100% refined grain (*n* = 9)	74 ± 15	55 ± 11	156 ± 45	137 ± 29	1.4 ± 1.1	1.8 ± 1.4	0.3 ± 0.2	0.3 ± 0.2	30 ± 9	25 ± 4
Multigrain mixed grain (*n* = 134)	75 ± 16	74 ± 11	191 ± 44	190 ± 31	2.6 ± 1.4	2.2 ± 0.8	0.4 ±	0.5 ± 0.2	35 ± 10	35 ± 6
Raisin bread 100% refined grain (*n* = 5)	63 ± 20	74 ± 12	206 ± 43	204 ± 40	2.9 ± 1.0	3.1 ± 0.7	0.8 ± 0.7	0.8 ± 0.6 *	40 ± 9	40 ± 7
Raisin bread mixed and or whole grain (*n* = 17)	75 ± 13	74 ± 14	199 ± 42	195 ± 43	3.7 ± 1.5	3.0 ± 1.7	0.6 ± 0.6	0.7 ± 0.6	37 ± 8	36 ± 8
	**Sugar (g)**		**Fibres (g)**		**Protein (g)**		**Sodium (mg)**		**Selling price (CA$)**	
	**Offer**	**Purchases**	**Offer**	**Purchases**	**Offer**	**Purchases**	**Offer**	**Purchases**	**Offer**	**Purchases**
Total (*n* = 294/*n* = 262)	3 ± 3	3 ± 2	3.8 ± 2.2	2.7 ± 1.7	7 ± 3	7 ± 2	289 ± 102	331 ± 83	0.59 ± 0.23	0.41 ± 0.10
100% whole grain (*n* = 39)	2 ± 1	3 ± 1	4.2 ± 2.7	4.2 ± 1.4 *	7 ± 2	7 ± 1	318 ± 90	355 ± 90	0.51 ± 0.31 *	0.38 ± 0.09
100% refined grain (*n* = 22)	3 ± 1	2 ± 1 *	1.7 ± 1.1 *	1.8 ± 0.4 *	6 ± 2	7 ± 1	306 ± 86	369 ± 83 *	0.42 ± 0.16 *	0.36 ± 0.06 *
Mixed Grain (*n* = 14)	2 ± 2	1 ± 1	4.1 ± 1.7	3.0 ± 0.9	7 ± 2	6 ± 1	309 ± 146	244 ± 95	0.57 ± 0.14	0.52 ± 0.10
Multigrain 100% whole grain (*n* = 54)	3 ± 2	2 ± 2	4.5 ± 1.8	5.5 ± 1.4 *	7 ± 3	9 ± 2 *	277 ± 112	288 ± 89	0.75 ± 0.25 *	0.49 ± 0.11 *
Multigrain 100% refined grain (*n* = 9)	1 ± 2	1 ± 1	1.7 ± 0.6	1.4 ± 0.5	5 ± 2	5 ± 1	244 ± 92	247 ± 22	0.54 ± 0.21	0.42 ± 0.12
Multigrain mixed grain (*n* = 134)	2 ± 2 *	3 ± 1	3.9 ± 2.1	2.5 ± 1.7	8 ± 2 *	7 ± 2	289 ± 102	320 ± 72	0.55 ± 0.17	0.41 ± 0.10
Raisin bread 100% refined grain (*n* = 5)	13 ± 6	15 ± 1 *	1.4 ± 0.9	1.4 ± 0.5	6 ± 1	6 ± 1	284 ± 76	271 ± 71	0.57 ± 0.09	0.60 ± 0.08 *
Raisin bread mixed and or whole grain (*n* = 17)	9 ± 4 *	7 ± 5	4.1 ± 1.8	3.3 ± 1.6	6 ± 3	7 ± 2	254 ± 61	261 ± 41	0.73 ± 0.22	0.52 ± 0.16

Mean ± standard deviation. Offer represents the average nutritional value of the bread offered on the shelves (*n* = 294). Purchase represents the average nutritional value of bread weighted by sales volume (*n* = 262). * Significantly, different from other bread 0.069% (*p* < 0.00069). This threshold equals the Bonferroni correction for offer and purchases separately (5%/72).

**Table 3 nutrients-13-04196-t003:** Nutritional value and selling price per two-slice portion of sliced bread offered and purchased in Quebec, according to the main flour.

	Weight for 2 Slices (g) or At Least 50 g		Energy (kcal)		Lipids (g)		Saturated Fats (g)		Carbohydrates (g)		Sugar (g)		Fibres (g)		Protein (g)		Sodium (mg)		Selling Price (CAD)	
	Offer	Purchases	Offer	Purchases	Offer	Purchases	Offer	Purchases	Offer	Purchases	Offer	Purchases	Offer	Purchases	Offer	Purchases	Offer	Purchases	Offer	Purchases
Main Flour																				
Whole wheat ^§^ (*n* = 103)	75 ± 15	75 ± 12	187 ± 42	183 ± 36	2.6 ± 1.3	2.6 ± 1.0	0.5 ± 0.3	0.6 ± 0.3	33 ± 8	33 ± 6	2 ± 2	2 ± 2	5.3 ± 2.2	4.9 ± 1.5	9 ± 2	8 ± 2	277 ± 98	305 ± 94	0.51 ± 0.14	0.44 ± 0.08
Refined wheat flour (*n* = 114)	71 ± 16	73 ± 11	189 ± 49	191 ± 33	2.4 ± 1.4	2.1 ± 0.6 *	0.4 ± 0.3	0.5 ± 0.2	36 ± 11	36 ± 7 *	3 ± 3	3 ± 3 *	2.4 ± 1.3 *	1.8 ± 0.8 *	7 ± 2 *	7 ± 1 *	302 ± 90	342 ± 76 *	0.48 ± 0.14	0.38 ± 0.09 *
Rice (*n* = 43)	68 ± 14	62 ± 11	181 ± 43	169 ± 36	5.1 ± 1.8 *	5.4 ± 1.7 *	0.7 ± 0.5	0.6 ± 0.4	34 ± 12	29 ± 7	4 ± 3 *	3 ± 2	3.4 ± 1.8 *	2.3 ± 1.8 *	4 ± 2 *	3 ± 1 *	276 ± 104	304 ± 64	0.95 ± 0.25 *	0.86 ± 0.10 *
Others (*n* = 34)	76 ± 13	76 ± 6	198 ± 45	197 ± 21	2.5 ± 2.3	2.8 ± 2.1	0.3 ± 0.2 *	0.3 ± 0.2	36 ± 8	35 ± 3	2 ± 2	2 ± 2	4.4 ± 1.8	2.9 ± 1.3	7 ± 2	8 ± 1	302 ± 138	358 ± 75	0.69 ± 0.17 *	0.62 ± 0.05

Mean ± standard deviation. Offer represents the average nutritional value of the types of bread found on the shelves (*n* = 294). Purchase represents the average nutritional value of bread weighted by sales volume (*n* = 262). * Indicates a significant difference at the threshold of 0.185% (*p* < 0.00185) between this type of sliced bread and the other types. This threshold corresponds to the Bonferroni correction (5%/27). ^§^ Reference category used for comparisons.

**Table 4 nutrients-13-04196-t004:** Multivariate analyses of the nutritional value and price per serving of bread purchased, based on the main flour.

	Energy (kcal)	Lipids (g)	Saturated Fats (g)	Carbohydrates (g)	Sugar (g)	Fibres (g)	Protein (g)	Sodium (mg)	Selling Price (CA$)
Main Flour									
Refined wheat (70.6%) **	5 ± 6	−0.7 ± 0.2 *	−0.1 ± 0.0	3.9 ± 1.2 *	1.5 ± 0.3 *	−3.2 ± 0.2 *	−1.8 ± 0.3 *	72 ± 15 *	−0.09 ± 0.01 *
Rice (0.7%)	−46 ± 107	−3.7 ± 2.6	−0.1 ± 0.8	−1.9 ± 20.4	−2.1 ± 4.4	−5.4 ± 3.6	−3.2 ± 4.3	65 ± 254	0.10 ± 0.23
Others (0.4%)	−16 ± 40	−1.2 ± 1.0	0 ± 0.3	−1.1 ± 7.6	0.0 ± 1.6	−4.0 ± 1.3	−1.6 ± 1.6	96 ± 94	−0.13 ± 0.08
Whole wheat ^§^ (28.3%)	0	0	0	0	0	0	0	0	0

Coefficient ± standard error. ^§^ Reference category used for comparisons. * Indicates a significant difference at the 0.555% threshold (*p* < 0.00555) between this category and the reference category. This threshold corresponds to the Bonferroni correction (5%/9). ** Percentages indicate the percentage of sales volume. The volume of sales, not the number of products, determines the power of the tests. The analyses in this table were adjusted for grain type, organic claim, natural or rustic mention, gluten-free claim, and market segments.

## Data Availability

The data presented in this study are available on request from the corresponding author. The data are not publicly available due to agreement signed with Protégez-Vous and ACNielsen.

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
