# Peer review of "Nutritional Content of Sliced Bread Available in Quebec, Canada: Focus on Sodium and Fibre Content"

_nutrients, 2021, doi:10.3390/nu13124196_

Round 1

Reviewer 1 Report

Thank you for the opportunity to review the article - Nutritional quality of sliced bread available in Quebec, Canada: focus on sodium and fibre content.

This study looked at the sodium and fibre content of bread and used sales data as an indicator of consumer consumption levels. An interesting study and valuable contribution to the literature. As the authors have highlighted, this study provides a baseline set of data for similar studies in the future to be compared against.

Title

The title refers to ‘nutritional quality of sliced bread’, yet the nutrient content (primarily sodium and fibre) is the only factor described in the paper. Consider the use of the term nutritional content or composition rather than nutritional quality – as the nutritional quality of a product is broader than just the nutrient composition (eg. Sliced bread vs artisanal bread).

Introduction

Line 30 – ‘Processed foods’ are mentioned in the first sentence and throughout the paper with very little definition provided. Given the increased interest in level of processing and use of terms including processed and ultra-processed foods, a clear definition of what is meant by processed foods in this paper is required. In the discussion, lines 452-454, the authors refer to less processed products being bread that is artisanal with fewer additives. This concept of bread having various levels of processing needs to be explained more clearly in the introduction for the reader.

Line 59 – ‘sliced bread’ – needs further definition here. After reading the entire paper, it was clear what was meant by the term sliced bread in this paper, but it would make it easier to read/ understand if this was explained in the introduction.

Line 109 – nutritional quality, should be replaced with nutritional content or composition. See line 163 where ‘content’ is used.

Materials and Methods

Lines 132-133 – this sentence clearly states the types of breads that are excluded from the study, but would have been good to see this information in more general terms provided in the introduction to help with the understanding of the definition of sliced bread.

Results

Line 212, Table 1.

  • The title of the table uses the word availability, yet in the text this concept is referred to as ‘offered’, elsewhere the word ‘supply’ is used. When reading the paper, these terms seem to all refer to the same thing so I wonder if one term can be used consistently throughout the paper to improve readability.
  • First line of table – formatting issue, suggest putting all information in brackets on one line.
  • Suggest ordering either availability or purchases data for type of grain and main flour from highest percentage to lowest to improve readability.

Line 233, Table 2.

  • Difficult to read second line of table. Suggest formatting supply and purchase as formatted in table 3 (text vertical rather than horizontal).

Line 242 – typo

Discussion

Line 342 – suggest replacing ‘consumption’ with ‘purchase’ – this will better align with the objectives in introduction.

Line 362/ 363 – consider replacing ‘previous study’ with ‘previous studies’ as it appears that two different studies are referenced here.

Line 446/ 447 – Explain what is meant by this sentence by providing an example.

Reviewer 2 Report

It is a very well written and interesting manuscript. 

In fact, only some minor format changes and adjustments are needed.

Line 12: unbold the word „Bread”

The Authors should adjust Table 2 because it is difficult to read at the current form. Maybe splitting into two tables would help.

Table 3 is also difficult to read. Consider swapping columns with rows in the table.
